# Predictors of successful return to parkrun for first-time adult participants in Scotland

**Andre S. Gilburn**＊

Healthy Environments: Sustainable Societies Research Group, Biological and Environmental Sciences, University of Stirling, Stirling, United Kingdom

＊ andre.gilburn@stir.ac.uk

## Abstract

Physical activity is essential for promoting good health and reducing burdens on healthcare systems. parkrun organise free weekly events where participants complete a 5km route. Studies have identified characteristics of participants associated with lower levels of participation. The aim of the study was to identify predictors of the likelihood of returning to parkrun for first-time adult participants. The return rate of adult first-time participants was determined for all 5km parkrun events in Scotland over a 1-year period from February 2019. The dataset consisted of 20,191 adult participants made up of 11,459 females and 8,732 males across 58 venues. A General Linear Mixed Model was used to identify factors associated with return rate. Return rates were negatively correlated with event size and positively correlated with the proportion of first-time adult participants at the event. Age was positively correlated with return rate and males were more likely to return. New participants that finished in a relatively slow time were disproportionately less likely to return. Return rates were positively correlated with the amount of freshwater and woodland on the route. These findings provide potential opportunities to manage events to enhance their efficacy. Specific events could be promoted as first-timer days to encourage new participants to attend together. New events could be prioritised in proximity to events that currently experience high attendances to reduce attendances locally. As the presence of freshwater and woodland are associated with higher return these habitats could play a role in generating the benefits of green exercise. If so the creation of more routes running through or alongside these habitats could be beneficial. The findings are likely to be widely applicable to other mass participation events and those interesting in understanding the mechanism by which green exercise provides its benefits.

## Introduction

Global patterns in physical activity

Estimates suggest there are 1.4 billion adults partaking in insufficient levels of physical activity globally [1]. This can lead to a range of diseases and place additional burdens upon healthcare systems [2]. Consequently, promotion of wider participation in physical activity has become a global priority [3]. The environmental context within which people exist can play an

**Data Availability Statement:** The full dataset is available in the University ofStirling datastorre http://hdl.handle.net/11667/210.

**Funding:** The author received no specific funding for this work.

**Competing interests:** The author has declared that no competing interests exist.

important role in driving levels of physical activity and defining patterns of inactivity [4]. These upstream factors can have both negative and positive influences and their management could influence their impacts. Upstream factors can include provision of more positive influences such as safe and pleasant physical spaces in which to exercise [4] and mass participation events that make use of these spaces to encourage group exercise and the additional social benefits that this generates [5].

## parkrun

One example of a positive upstream factor is parkrun who organise approximately 2000 free weekly 5km events across 22 countries [6]. These events often occur in public parks but also make use of other suitable public areas. Events also occur on private property such as estates, university campuses, nature reserves and forestry. Events are designed to promote inclusivity with participants allowed to use wheelchairs, run with young children, use walking frames, push buggies and run with a dog. There is no also requirement to complete the route within a time limit with walking now being actively promoted through the parkwalk at parkrun initiative [7].

parkrun have a mission to create a healthier and happier planet [8]. Studies into the impacts of parkrun have revealed that it does indeed provide both a physical and mental health benefit to its participants [9–13]. A combination of initiatives and the inclusive regulations means that some traditional barriers to partaking in sport and exercise are not present within parkrun [5]. Studies have also shown there are more barriers to participating in physical activity for women than men [14, 15] and that the context in which women will engage in physical activity can differ from that of men [16, 17]. Motivations for taking part in physical activity also differ between the genders [18, 19]. Understanding how the genders respond to the provision of positive upstream factors, such as parkrun, is likely to be crucial to their management and effectiveness [20].

In addition to physical health benefits, both participants and volunteers gain social network rewards from being part of the parkrun community [5, 21–28]. Engagement with parkrun has also been found to promote a more positive sense-of-self [29]. Testament to the success of parkrun is highlighted by the important role it is now playing in social prescription, with many medical practices in the UK being linked to specific parkrun locations, with parkrun being prescribed to patients as part of their treatment [30–33]. So, the benefits of engaging with parkrun are truly multifaceted.

## Green exercise and parkrun

As all parkrun events occur outdoors they encourage engagement with the natural environment with studies identifying additional benefits of green exercise [5, 34–37] compared to exercising indoors in a gym. Studies have also found that exercising in more natural environments seems to have more benefits compared to exercising in urban green spaces [36]. parkrun routes utilise both urban green spaces and other more natural environments. The setting could influence the benefits gained from attending a parkrun. For example, completing a parkrun event that runs alongside water might also provide additional mental health benefits as exposure to blue spaces has been found to be associated with better mental health [38].

## Using data science to further our understanding of parkrun

The majority of studies of participation in parkrun have used qualitative methodologies particularly in the form of surveys. Quantitative approaches such as data science can also be applied to identify predictors of engagement with parkrun by through large-scale statistical modelling

of the parkrun results dataset. Data science can be used to address specific questions about participation in parkrun and also to identify previously unknown patterns and associations between various factors and participation levels [10, 22, 39, 40]. One key advantage of data science studies is the samples will not be affected by survey bias [41]. When participants register with parkrun they are allocated a unique identification number [39]. They use this number after completing a parkrun to get their finishing time and be included in the results. The results of each event are published on the parkrun website generating a dataset on the finishing time, gender and age group for millions of participants and many millions of participations. The dataset could contain considerable amounts of novel information about associations between levels of participation and various characteristics of both participants and events. There have been relatively few quantitative studies utilising the results of parkrun events to further our understanding of its impacts and benefits [10, 22, 39, 40]. This relative lack of studies utilising the parkrun results database is surprising as one of the studies that pioneered research into parkrun used finishing times as a proxy for fitness [10].

Three studies have used the parkrun dataset to understand patterns of participation [22, 39, 40]. The first study was conducted in Tasmania and found that those with lower levels of education are more likely to regularly participate in parkrun showing evidence of how inclusive it can be [22]. A study in Scotland found that the performance of the parkrun population was falling even though it was on average increasing for individual participants showing that parkrun was becoming increasingly inclusive by attracting more inactive participants [39]. A recent study in the UK, Ireland and Australia has identified that 43% of people who register for parkrun never take part in an event and a further 22% only participate once [40]. This study questioned those that registered for parkrun to identify barriers to participation and returning. This identified that women, younger adults and the inactive were least likely to participate or return. The main barriers identified were the inconvenient start time and the feeling of being too unfit to participate. The latter was more commonly reported by women than men.

## Aim of the study

There remain many unanswered questions with respect to what motivates people to engage with parkrun. The overarching aim of this study is to use quantitative analyses of the behaviour of new parkrun participants to address some of these key knowledge gaps and to identify previously unknown associations with returning to participate in parkrun for a second time. The questions addressed by the study can be separated into two broad groups.

1. What characteristics of new participants predict their likelihood of returning to parkrun? Previous studies have investigated associations between age and gender with return rates. This study includes an additional new characteristic of participants, their finishing time, and investigates previously unexamined interactions between these factors.

2. What characteristics of parkrun events predict the likelihood of new participants returning to parkrun? The following characteristics of events were considered: field size; proportion of new participants; gender ratio of participants, the proportion of land cover types the routes proceed through; distance to the next nearest event; elevation gain; surface type. No previous studies have investigated if any of these characteristics are associated with levels of participation in parkrun, however, one previous study has shown that distance to the next nearest event, elevation gain and surface type are all associated with performance [39] which another has shown to be associated with levels of participation [10].

These overarching research questions will enable the following key specific hypotheses to be tested for the first time.

1. Is the proportion of other new participants at a parkrun event associated with the likelihood of new participants returning to parkrun?

2. Is the proportion of other new participants at a parkrun event associated with the likelihood of new participants returning to parkrun?

3. Is the finishing time of new participants at a parkrun event associated with likelihood of them returning to parkrun?

4. Is the land cover type that a parkrun route passes through associated with likelihood of new participants returning to parkrun?

5. How do the gender and age of participants interact with any new associations identified when answering the specific questions above?

## Methods

### Ethics statement

This was an analytical study of aggregated secondary data and as such had no active participants. Ethical approval was obtained from the Stirling University General University Ethics Committee (EC 2022 10861 8035).

### parkrun data

The study included data from all 58 5km parkrun course locations that held a parkrun event in Scotland over a year long period from February 2019 to January 2020. The course locations comprised a range of event location types including 33 in public parks, eight coastal events using esplanades and the areas surrounding them, eight events used traffic-free paths, three occurred on private estates, three on university campuses, one in a forestry, one on a Local Nature Reserve and one on public roads.

The results page for all events were accessed and processed using an Excel macro which extracted information about each participant including their age category, parkrun ID number, gender, age group, finishing time, number of participations, date and whether the participant was a new parkrunner [42]. An example results page is available here [43]. Any unknown participants (participants who completed the event without presenting an identification number) and participants under 18 years of age were removed from the dataset. Adults participating in their first parkrun were identified amongst the remaining participants. All other records were removed resulting in a dataset of new parkrun participants who attended an event in Scotland over that one-year period. The dataset consisted of 20,191 adult participants made up of 11,459 females and 8,732 males across 58 different event venues.

Age for adult participants is provided in the parkrun results as a 5-year cohort except for 18–19 year olds. Age was converted to a continuous variable by assigning participants the mid-point for their cohort group for all new parkrun participants. The parkrun identification number of all new participants was used to access their parkrun participation history to determine whether they had subsequently returned to parkrun. These were accessed in November and December 2022. Consequently, new participants had a period of at least 33 months up to a maximum of 46 months after their first participation to return to parkrun. It should be noted that Scottish parkrun events were suspended for a period of 17 months between March 2020 and July 2021 so the return time was in effect a shorter period of 16 to 29 months where active events were occurring in Scotland.

Additional characteristics were collected for the first event each of the participants attended. These were the number of participants, the number of new adult participants and the gender ratio of the participants. The following additional event characteristics were used: elevation gain in m; surface type (scored 0 for soft surfaces such as trail, 1 for mixed soft and hard surfaces and 2 for hard surfaces such as tarmac); the shortest travelling time in minutes by car from the recommended parking of an event to the recommended parking of another parkrun events as determined from Google Maps. This was used as a measure of the remoteness of an event from other parkruns [39].

## Land cover data

Each parkrun event location provides a map of its route. An example is available here [44]. The course routes for all 58 Scottish event venues that hosted a parkrun during the study period were downloaded in Keyhole Markup Language format and imported into the GIS software package Digimap Edina [45]. Measuring tools within the aerial roam feature were used to mark out a 30m distance from the route and the total area within the zone determined and recorded. The land cover types within the zones were classified using satellite imagery on Digimap and the proportion of each type within each zone determined. Land cover was classified into the following types: woodland, grassland, freshwater, saltwater, shore, urban and other. Other included anything that could not be classified into the other six land cover types and only comprised on average just 2.3% of the total area surrounding events.

## Statistical methodology

The data were analysed using R x64 4.1.1 [46]. A generalized linear mixed model (GLMM) with a binomial error distribution was used to model participants returning to parkrun. This was generated using the glmer function in the lme4 package [47]. Quadratic terms were included in the model of returning to parkrun for finishing time, number of participants, proportion of first-time participants and date. Event venue was included as a random effect. All continuous explanatory variables were scaled to have a mean of zero and a standard deviation of one including quadratic terms. Minimum Akaike Information Criterion was used to select the optimal model.

## Results

### Factors determining the return rate of first-time parkrun participants

The overall return rate of first-time participants to parkrun was 64.18% (12,959 of 20,191). A GLMM identified several significant associations with return rate (Table 1). There was a significant increase in return rate with age (Table 1, Fig 1). Date was strongly negatively associated with return rate. A significant quadratic term shows that the association with date weakens over time. Male participants (66.5%, 5,805 of 8,732) were highly significantly more likely to return (Fig 1) than female participants (62.4%, 7,154 of 11,459). The finishing time at a participants first parkrun was also an important determinant of return rate. The non-significant linear term and significant quadratic term suggests little effect of time on return amongst the faster runners but a disproportionately negative impact of the slowest times on return rates (Fig 2). There was a highly significant quadratic association between the proportion of new participants at an event and return rate, with participants disproportionately more likely to return after attending events with the highest proportion of new participants (Fig 3). There was also a highly significant association between event size and return rate with first-time participants more likely to return when attending smaller events. The travelling time to the next

**Table 1. A GLMM with binomial error distribution of return rate to parkrun of adult first-time participants.** All continuous explanatory variables were scaled. Three interaction terms and three quadratic terms were retained in the model.

| Parameter | $Z_{20,174}$ | Estimate | Standard Error | P |
|---|---|---|---|---|
| Intercept | 15.09 | 0.556 | 0.036 | <0.001 |
| Age | 9.97 | 0.159 | 0.016 | <0.001 |
| Date | 3.99 | -0.058 | 0.015 | <0.001 |
| Date$^2$ | 3.98 | 0.058 | 0.015 | <0.001 |
| Gender(male) | 2.40 | 0.083 | 0.034 | 0.016 |
| Proportion of first-timers | 1.81 | -0.075 | 0.042 | 0.070 |
| Proportion of first-timers$^2$ | 3.59 | 0.155 | 0.043 | <0.001 |
| Number of participants | 3.39 | -0.103 | 0.030 | 0.001 |
| Finishing time | 1.26 | 0.117 | 0.093 | 0.207 |
| Finishing time$^2$ | 2.95 | -0.266 | 0.090 | 0.003 |
| Travelling time to next parkrun | 2.69 | -0.067 | 0.027 | 0.012 |
| Surface type | 1.77 | 0.058 | 0.033 | 0.077 |
| Woodland | 2.34 | 0.090 | 0.038 | 0.019 |
| Freshwater | 2.31 | 0.092 | 0.040 | 0.021 |
| Woodland*Freshwater | 1.51 | 0.078 | 0.052 | 0.131 |
| Gender(Male)*Freshwater | 2.54 | -0.076 | 0.030 | 0.011 |
| Age*Freshwater | 1.92 | 0.029 | 0.015 | 0.055 |

nearest parkrun also was negatively correlated with return rate. The mean travelling time to the next nearest event was 30 mins for those that returned and 33 mins for those that did not return.

Return rates were also positively associated with the amount of woodland and freshwater at an event. In the case of freshwater, a significant interaction term with gender reveals a stronger association between return rate and the amount of freshwater on a route in female than male participants. Gender ratio of the field and elevation gain were not retained in the model.

## Discussion

### What characteristics of new participants are associated with their likelihood of returning to parkrun?

The overall return rate of 64.18% is higher than the 61.95% reported in an earlier study [40]. In the current study participants had a longer period of time within which to return (33–46 months compared to 2 years). However, Scottish parkruns were suspended for 17 months during the pandemic meaning that the possible return period was effectively 16–29 months which is on average slightly shorter than the previous study. This suggests that Scottish parkruns might have a relatively high proportion of new participants that return to parkrun. As this study included data from both sides of lockdown an additional study of the impact of lockdown on the return rates of new parkrun participants would be needed to fully understand if Scotland does on average have higher return rates than the rest of the UK, Ireland and Australia. Furthermore, another study identified that parkrun participants in Scotland are increasing in age [39], so this could simply be a consequence of an older population of parkrun participants being considered here compared the previous study, as older new participants are known to be more likely to return.

Age and gender were both found to affect likelihood of returning to parkrun for first-time participants in Scotland. This is consistent with the finding of the larger scale study covering

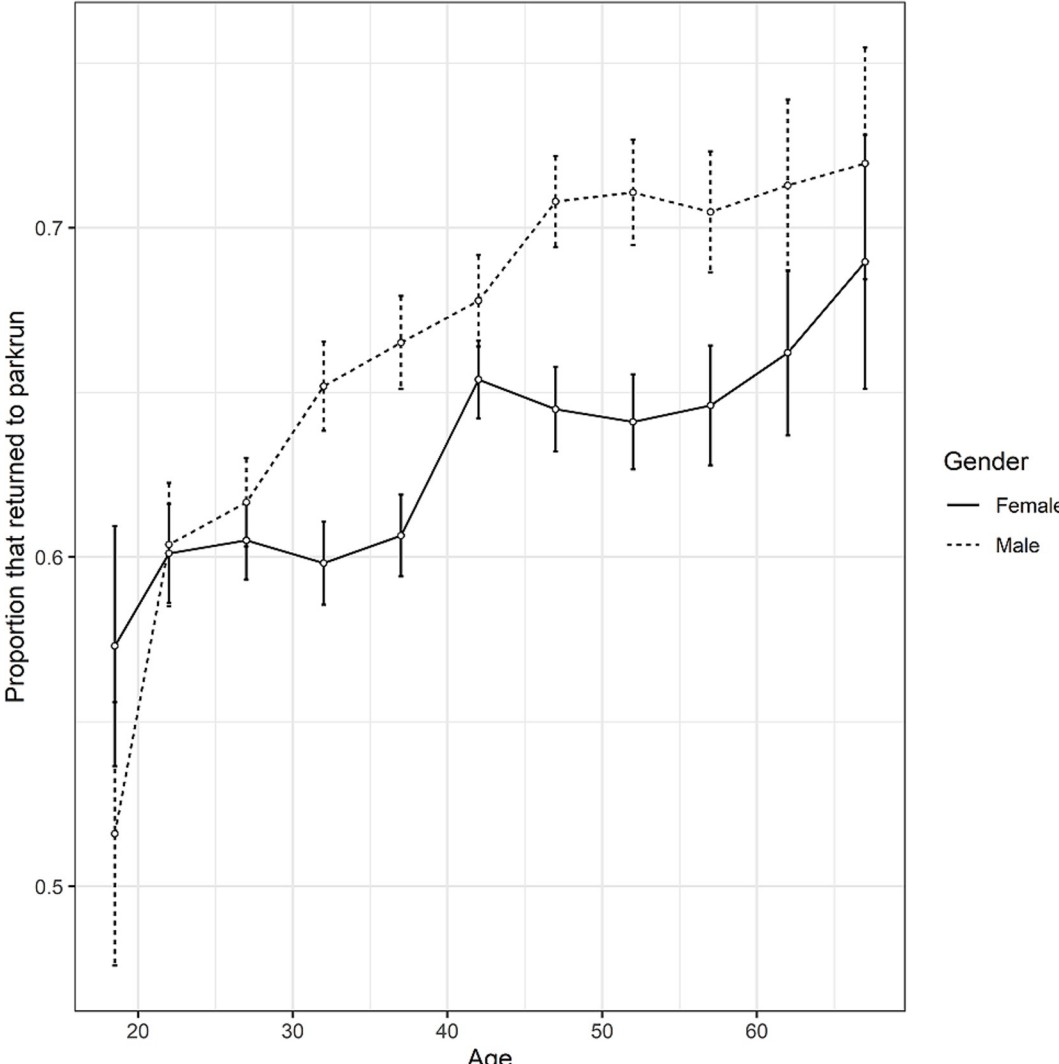

**Fig 1. The return rate of adult first-time participants to parkrun events in Scotland against their age.** Standard error bars are provided for the mean return rate for each age cohort for both sexes.

the UK, Ireland and Australia [40]. The gap in return rate between the genders was slightly wider in Scotland (66.5% for males, 62.4% for females) compared to the broader geographic study that found return rates of 63.7% for men and 60.4% in women [40]. The wider gap in Scotland might be indicative of a larger gender gap in activity in Scotland compared to other regions. A gender gap in activity has been previously reported among school children in Scotland [48]. Extending the current study design to include other regions would allow determination of whether Scotland does have a relatively wider gender gap compared to some other parkrun nations.

The broader geographic study found from surveying participants that those that engaged more readily in physical activity were more likely to return. The current study found that first-time participants with finishing times of over 40 minutes (Fig 2) had disproportionately low return rates. Those that exercise more regularly are likely to be fitter, and therefore run faster times [10] so this finding is also consistent with the previous study suggesting that the least active, and those most likely to benefit from parkrun, are those that are least likely to return [40].

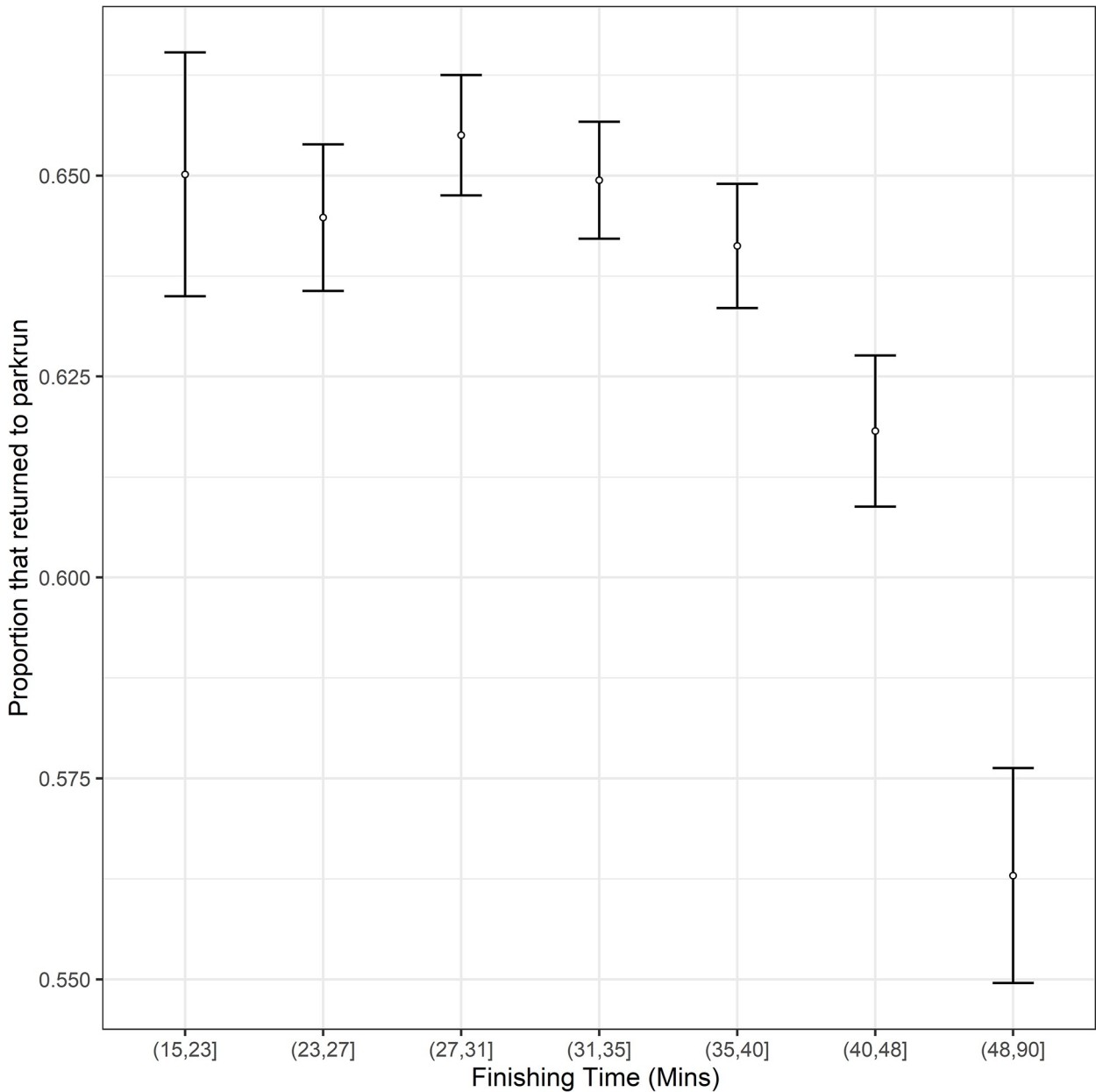

**Fig 2. The return rate of adult first-time participants to parkrun against their finishing time.** Standard error bars are provided for cohorts based upon finishing time. N.B. Finishing time was treated as a continuous variable in the analyses. Cohorts have simply been created to aid illustration of the data.

A key barrier to participation has been identified in a previous study as the psychological fear of not being fit enough to participate [40]. The current study found that finishing with a particularly slow time was associated with lower return rates which adds additional evidence that being relatively unfit is a barrier to continued participation. The parkwalk at parkrun initiative was introduced in September 2022 with the aim of encouraging and promoting walking. This could really help make slower new participants feel more welcome. A study looking at return rate of relatively slow new participants before and after parkwalk at parkrun was

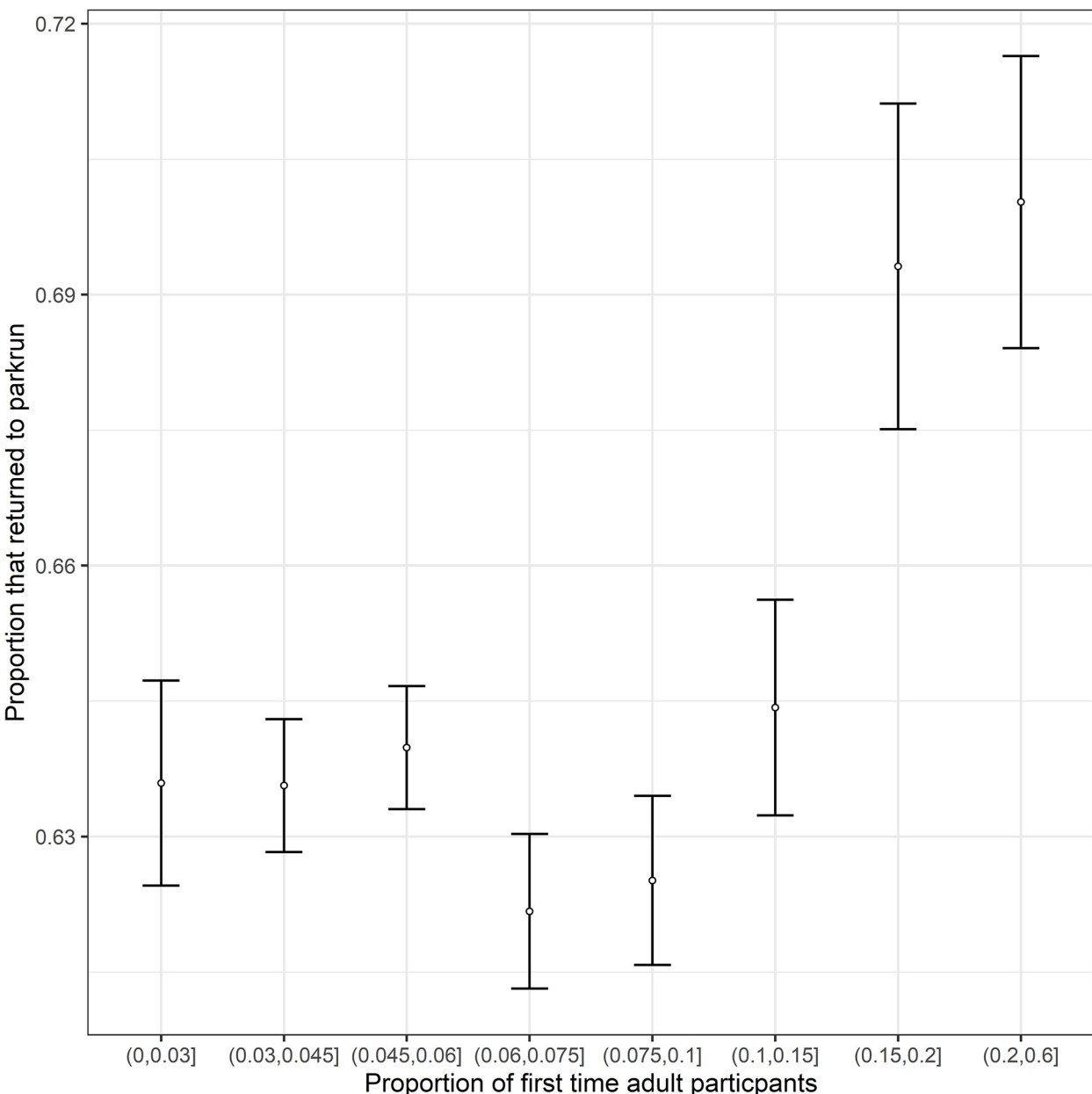

**Fig 3. The proportion of first-time adult participants to Scottish events that returned to parkrun against the proportion of new adult participants at the event venues they attended.** Standard error bars are provided for cohorts based upon the proportion of adult participants attending events. N.B. Proportion of new adult participants was treated as a continuous variable in the analyses. Cohorts have simply been created to aid illustration of the data.

launched could determine if it is contributing to making slower new participants feel more welcome by increasing the proportion of them that return.

## What characteristics of parkrun events are associated with the likelihood of new participants returning?

This study was the first to investigate what characteristics of parkrun events are associated with the return rate of first-time parkrun participants. Those who attended events with a

higher proportion of other adult first-time parkrun participants were more likely to return. First-time parkrun participants were also more likely to return after attending events with smaller field sizes. These two findings, together with the finding discussed earlier that new participants recording particularly slow finishing times were particularly unlikely to return, could relate to how much a new parkrun participant feels a part of the parkrun community. The social rewards gained from feeling part of the parkrun community have been identified as being important in a range of studies [25–27, 35, 49, 50]. Attending events with friends and family has been found to be an important motivator for participating in parkrun [35]. This shared experience of completing their first run with others could be contributing to them feeling part of the parkrun community. The higher return rate of those attending with other new participants could be partly driven by new participants who are known to each other attending their first event together. Future studies investigating whether it is individuals attending their first events in groups or unknown individuals sharing the experience of their first participation that drive this association would be useful for understanding the management implications of this finding.

First-time participants who attended an event where the travelling time to the next nearest event was shorter were also more likely to return. This suggests that density of parkrun event locations within an area could be influencing return rates with individuals more likely to return to parkrun in areas with a higher density of event venues.

This study was only the second to try to relate geospatial features of parkrun routes to a measure of the outcome of the experience, in this case likelihood of returning to parkrun. Both the amount of freshwater and woodland that new parkrun participants were exposed to were positively associated with their likelihood of returning. Studies have revealed additional benefits to green exercise in more wild landscapes than urban green spaces [36]. Assuming return rates of new participants to parkrun are related to level of positivity of the experience then the current study suggests in the context of parkrun that woodland and freshwater could be encouraging people to return by enhancing the quality of green exercise. This would mean there is potential to manage the influence of parkrun as a positive upstream factor by prioritising the creation of new routes alongside freshwater and woodland.

Studies investigating what aspects of exercising in woodland and alongside freshwater might enhance the green exercise experience would be valuable. For example, what is the nature of the stimuli enhancing the benefits of exercising in green spaces. Is it visual stimuli from for example seeing trees and water. Is it olfactory stimuli, for example smelling plants such as wild garlic in woodlands, or it is auditory stimuli such as the sounds of bird calls. The fact that parkrun have numerous events all taking place at the same time and on the same time of the week but in varying locations means that parkrun provides a potentially unrivalled opportunity for understanding how the benefits of green exercise are gained.

Another notable finding from the study was that freshwater was positively associated with return rates but saltwater was not. This could suggest that freshwater has a more positive impact upon parkrun participants but alternatively it could be caused by another correlated factor. For example, coastal areas will experience higher wind speeds, which might have a negative impact on return rates and possibly negate the positive benefits gained from running alongside water. Further study is required to establish why the experiences of running alongside freshwater and saltwater has different impacts on return rates of new parkrun participants and what relevance this might have to the benefits of green exercise.

## Implications for parkrun

One of the key findings of this study is that in terms of encouraging continued engagement with parkrun not all parkrun venues are equal. Variation in size, distance from other events,

proportion of new participants and the amount of woodland and freshwater on the route are all associated with return rates. Therefore, it might be possible to enhance the role of parkrun as a positive upstream factor by increasing the attractiveness of parkrun venues to new participants. As route design could influence whether first-timers return, parkrun might want to consider the mechanism by which new events are created and whether they can add more flexibility to these processes to encourage the introduction of more parkruns with characteristics associated with higher return rates. For example, parkrun have prioritised the creation of events on socio-economic grounds after inequalities were identified in the distribution of parkrun events in England [51]. After receiving funding to create 200 new events a study identified the ideal locations to reduce these inequalities [52]. In addition to more managed event creation, the fact that this study identified that return rates were higher at smaller events and at events where the travelling time to other events was lower, both support the continued creation of more parkrun venues through more traditional routes, assuming that local communities can sustain additional events by providing enough volunteers willing to contribute to organising them.

The discovery that new participants are more likely to return after attending an event with a higher proportion of other first-time participants suggests that parkrun might want to consider introducing specific event days to which new participants are particularly encouraged to attend, so increasing the proportion of new attendees present at those events.

Another implication of the findings of this study relates to the parkrun practice initiative [30–33]. Practitioners utilising the parkrun practice might want to consider prescribing specific local event parkrun events that could particularly increase the patient's likelihood of returning, for example, by recommending smaller events or where the route goes through woodland and alongside freshwater. Furthermore, patients could be encouraged to attend their first parkrun as a group, or with friends and family.

## Limitations of the study

Data science studies are excellent for identifying previously unknown associations between variables but limited in their ability to determine cause and effect as the data are not primarily collected in order to test predictions of hypotheses. This means that all the findings of the study are correlational. Consequently, data science studies of this type are particularly useful for generating new hypotheses that can be tested in other studies but are more limited in providing tests of those hypotheses.

The study was also limited to a year-long period due to the considerable time needed to generate the dataset. It is known that the gender gap in participation has been narrowing and this could be partly driven by changes in the return rate of the different genders over time [39]. Therefore, it would be useful to compare this study to other time periods to determine how general the findings are. It is notable in this study that although return rates were lower for female first-time participants there were still more returning female than male first-time participants because the difference in return rate was more than compensated by the higher proportion of female first-time participants. Therefore, the study period was associated with a clear narrowing of the gender gap and shows that, at least in Scotland, the majority of new participants that return to parkrun are actually female. This suggests that parkrun is successfully overcoming traditional barriers to female participation in physical activity.

The restriction of the study to Scotland is another limitation. It would be useful to conduct similar analyses of parkrun return rates in other areas to determine the generality of the findings.

The COVID-19 pandemic resulted in parkrun being suspended in Scotland for a period of 17 months. This could have impacted the return rates of participants as lockdown is known to

have substantially impacted the levels of physical activity with reductions in activity particularly amongst the less fit, the young and women [53]. The sample duration for first-time participants was specifically chosen to cover the full one-year period before news of the potential pandemic hit the media to reduce the impact of the pandemic influencing the sample but while still making the study reasonably current. Therefore, all participants included in this study attended their first event without knowledge of the pandemic, but many will have not returned by the point that news of the emerging pandemic was hitting the media. The higher return rates found in this study which straddles the covid suspension compared to the one previous study [40] which was conducted prior to the pandemic suggests that it may not have substantially negatively impacted return rates. It would be interesting to investigate the impacts of the pandemic on return rates further by investigating them for first-time participants from February 2020 onwards when they would have known about the pandemic when they first attended parkrun.

## Conclusion

This study has identified various novel features of parkrun events that are associated with the likelihood of first-time participants returning to parkrun. Identification of these features provides parkrun with additional information that could be potentially used to manage their events to increase their efficacy. The results also have potentially important wider implications for other organisers of mass participation events as the same characteristics associated with return rates at parkrun events are likely to be more widely applicable. The findings of the study also extend our understanding of green exercise by suggesting that exercising in woodland and alongside freshwater could be important components of the benefits gained by exercising outdoors in a green space.

## Acknowledgments

The author would like to thank Euan McDiarmid for generating the landcover data. The author would also like to thank Kate Black and an anonymous referee for providing constructive comments on an earlier version of the manuscript. The author acknowledges the use of data owned by parkrun Global. The data have been accessed as a permitted act for independent non-commercial research purposes through fair dealing legislation allowing access to publicly available databases. Only a tiny proportion of the parkrun results database was accessed (data from just 58 of more than 2000 events). The author has no official connection to parkrun but is a regular participant and volunteer.

## Author Contributions

**Conceptualization:** Andre S. Gilburn.

**Data curation:** Andre S. Gilburn.

**Formal analysis:** Andre S. Gilburn.

**Investigation:** Andre S. Gilburn.

**Methodology:** Andre S. Gilburn.

**Writing – original draft:** Andre S. Gilburn.

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
