## [Decision Letter · Decision Letter 0]

25 Apr 2023

PGPH-D-23-00368

Characteristics of parkrun That Encourage New Participants to Return

Dear Dr. Gilburn,

Thank you for submitting your manuscript to PLOS Global Public Health. After careful consideration, we feel that it has merit but does not fully meet PLOS Global Public Health’s publication criteria as it currently stands. Therefore, we invite you to submit a revised version of the manuscript that addresses the points raised during the review process.

Your manuscript has been assessed by two expert reviewers, whose comments are appended below. The reviewers have highlighted concerns about several aspects of the methodology and the clarity of the rationale for the specific study design you used. Please ensure you respond to each point carefully in your response to reviewers document, and modify your manuscript accordingly.

We look forward to receiving your revised manuscript.

Kind regards,

Dr Joseph Donlan

Senior Editor

Journal Requirements:

1. Please amend your online Financial Disclosure statement. If you did not receive any funding for this study, please simply state: “The authors received no specific funding for this work.”

2. Please update your online Competing Interests statement. If you have no competing interests to declare, please state: “The authors have declared that no competing interests exist.”

3. Please provide separate main figure files in .tif or .eps format only and ensure that all files are under our size limit of 10MB.

4. We have noticed that you have uploaded Supporting Information files, but you have not included a list of legends. Please add a full list of legends for your Supporting Information files after the references list.

Additional Editor Comments (if provided):

Reviewers' comments:

Reviewer's Responses to Questions

**Comments to the Author**

1. Does this manuscript meet PLOS Global Public Health’s publication criteria? Is the manuscript technically sound, and do the data support the conclusions? The manuscript must describe methodologically and ethically rigorous research with conclusions that are appropriately drawn based on the data presented.

Reviewer #1: Partly

Reviewer #2: Yes

2. Has the statistical analysis been performed appropriately and rigorously?

Reviewer #1: Yes

Reviewer #2: I don't know

3. Have the authors made all data underlying the findings in their manuscript fully available (please refer to the Data Availability Statement at the start of the manuscript PDF file)?

Reviewer #1: No

Reviewer #2: Yes

4. Is the manuscript presented in an intelligible fashion and written in standard English?

Reviewer #1: Yes

Reviewer #2: Yes

5. Review Comments to the Author

Reviewer #1: Thank you very much indeed for the opportunity to review your article on the characteristics of parkrun that encourage new participants to return. Your article has used linear modelling and associated statistics on pre-existing data (parkrun results database) to assert that return was more likely for older males, faster runners, smaller events, and where there were a greater number of other first timers.

As also holding the identity of a Run Director for parkrun in an area of high obesity and a very broad socio-economic status (but a large event without water on the route itself!), I found this especially interesting. Your use of pre-existing data from across parkrun events is a very suitable and objective dataset which reduces potential issues around for example perceptions of returning. I do though have concerns that if this paper was read by someone outside of the community that they might struggle to understand what parkrun is, how it operates and what this data you are using is.

I do also have concerns about the data used and the decisions that you have made at the outset, without an extensive review of the literature, notably, that only certain characteristics will encourage return. In consequence, it is risky asserting that you are identifying “characteristics” of parkrun that encourage return. You are actually identifying a subset of characteristics from a very bounded set of characteristics.

Overall, I think this paper has potential to contribute to the field, but does need significant reconsideration from the current format.

Consequent of these overall concerns, I do have significant reservations about the paper. My major concerns are as follows:

• The structure of the Introduction is unclear. For example, it is unclear what parkrun is and why it is important. At present, the structure of the paper assumes insider knowledge. For example, where are these events located (e.g. line 65 needs context!). I think that the paper needs to start with an introduction to the growth in physical activity before moving to the barriers that individuals might face. You should also provide some explanation or evidence as to why you are indicating a connection between women and morbid obesity. I actually don’t think that you are suggesting this but the construction of the first paragraph indicates that this is the case!

• No clear aim of the paper and/or research question is stated. You merely assert that you are applying a generalized linear model to the data – but why is this necessary?

• You have not really examined the literature to document what previous studies both within parkrun but also relating to exercise more broadly, have shown / told us. You assert that previous studies of parkrun have used descriptive statistics. This omits to recognise that qualitative studies have also been undertaken, for example, Warhurst & Black (2022) “Lost and found: parkrun, work and identity”, Qualitative Research in Sport, Exercise and Health https://doi.org/10.1080/2159676X.2021.1924244

• As indicated above, you have discerned only a very limited number of characteristics that you consider will encourage return without justification for the selection of these specifically. An extensive literature within health, psychology and sport/exercise has examined multiple other factors that have been shown to influence likelihood of exercise (and of parkrunning). For example, why have you collected data re water without considering other terrain that have been shown to enhance wellbeing such as open space generally within large urban environments (e.g Birch, J., Rishbeth, C., & Payne, S. R. (2020). Nature doesn't judge you–how urban nature supports young people's mental health and wellbeing in a diverse UK city. Health & Place, 62, 102296; also Martin, L., White, M. P., Hunt, A., Richardson, M., Pahl, S., & Burt, J. (2020). Nature contact, nature connectedness and associations with health, wellbeing and pro-environmental behaviours. Journal of Environmental Psychology, 68, 101389). There are a number of systematic reviews of this literature. In the case of parkrun itself you have also not taken account of what has been shown to be a key factor in sustaining the parkrun community – that of the social aspects that surround the run such as the availability of a café or refreshments and/or post-run community gathering/s – see for example, Hindley, D. (2022). “More than just a run in the park”: an exploration of parkrun as a shared leisure space. Leisure Sciences, 42(1), 85-105; also Morris, P., & Scott, H. (2019). Not just a run in the park: a qualitative exploration of parkrun and mental health. Advances in mental health, 17(2), 110-123

• Methods: It is not clear from where you sourced the data that you analysed? You indicate that it is the parkrun results but someone without knowledge of parkrun won’t know what/where. Why only 56 events that data was extracted from? Why those 56? Were these just UK? Have you taken account of location more broadly?

• Within the findings, a number of assertions are made that are unclear:

o (line 183/184) that “those that engaged more readily in physical activity were more likely to return” but what is meant by this as you do not have data on exercise readiness?

o (lines 198-200) “Individuals who enjoyed the general experience of parkrun but maybe not the specific course at the venue they attended could be more likely to try a different parkrun if they have another local alternative”. What is your evidence for this assertion?

o (lines 203-204) “first-time participants with the slowest finishing times were more likely to attend smaller and more remote events”. Do you know this, or is it that smaller events typically have slower participants? – and perhaps there is a reason for this, for example, are they new events, remote events, specific terrain etc. etc.?

• You do not interpret the data/findings is made. As one example, the first para suggests, that returning to Scottish parkruns had a higher likelihood than shown in the wider data but it isn’t clear why (needs explanation) and the implications of this – especially given the health data for Scotland as a whole (although whether the hard-to-get-to are actually ever parkrunning …??)

• The implications asserted from the findings indicate that the author does not fully understand the context of parkrun and how/when new parkruns are established! The establishment of a parkrun depends upon the availability of an open space that facilitates a safe 5k route, the availability of a team of volunteers (typically relates to the socio-economic status of the area / social capital of the lead), the willingness of a landowner. New concepts / practices are asserted at this stage. For example, social prescribing is potentially really important for the whole paper.

In addition to the more major concerns that I have, I feel that greater clarity is needed as follows:

• p.3 It needs to be clear that parkrun is a running event as the preceding sentence refers to cycle routes.

• Further explanation of parkrun is also needed as the current description assumes insider knowledge of the events and the datasets. For example, do you mean that by 2020 over 30m runs had been completed? (and is the COVID pandemic actually a necessary statement?).

• Ensure that all assertions are evidenced – for example, line 68, previous studies – what previous studies?

• Take care using the term “primary data source” (line 82) as this might lead the reader to think you are generating primary data (ie an empirical study)

I look forward to seeing a revised version of the paper.

Reviewer #2: Review

This paper presents secondary data analysis from the parkrun database. It focuses on adult parkrun participants in Scotland between 2019-2020 – looking particularly at new participants who return. The authors have presented a generalised linear mixed modelling analysis and found important factors that influence whether participants return. These factors are important considerations for parkrun in encouraging new participants to return and when planning new events. This is a well-written manuscript; the text is clear and easy to read. It is also relevant to the journal and would interest the reader. My main concern with the paper is in level of detail. A stronger rationale is needed for why the paper focuses on parkrun, more detail is needed in the methods to explain how “remoteness” and travelling time were determined and the wider implications of the findings (beyond direct implications for parkrun) could be considered as this will interest a wider readership.

Major issues with this paper:

1. I would like to see a stronger rationale/justification for why parkrun is being used as an example of a “positive upstream factor” in this paper. What role could parkrun play in the global obesity epidemic? – perhaps here draw upon the potential for parkrun as a social prescription offer by health practitioners. Is it just the obesity epidemic that parkrun is potential useful for? Why is it important to understand patterns of participation, what could we do with this information? Perhaps the authors could use the existing parkrun research evidence base more to support this justification.

2. The methods section would benefit from detail around:

a. How remoteness of parkrun venues was determined

b. How travelling time to parkrun venues was determined

3. Parkrun is used here as an example of a ‘positive upstream’ initiative that can have an important role in the obesity epidemic.

Minor issues

Please see below for issues related to each manuscript section.

Title

I think the title would benefit from extra detail about the methods and/or location, also that the focus is on adults - if title word limit allows. A suggestion below – just a suggestion!

Characteristics of parkrun That Encourage New Adult Participants to Return: Generalised linear mixed modelling of parkrun data from Scotland

Abstract

Abstract provides a succinct summary of the research. More detail of the methods would be beneficial. For example – how many parkrun events were analysed

Line 16: GLMM is used in the abstract – should be written in full.

Introduction

General comments about Introduction: The introduction mentions the obesity epidemic, and parkrun’s role as a ‘positive upstream factor’ in this. Doing so might be reducing parkrun to a physical activity intervention only, when the research suggests it could be more than that (e.g., community initiative, social capital, wellbeing, volunteering). Is the obesity epidemic the only public health concern that parkrun could address?

Line 31-37: The first paragraph mentions physical activity and sport – with terms seemingly being used interchangeably. Is this intentional? It could be clear – perhaps with a definition of physical activity that encompasses sport and exercise.

Line 35: 1.4 billion – is that globally?

Line 45-46: could add that the 2,000 events are across 22 countries.

Line 49: “covid-19 pandemic” is later referred to as “SARS-cov2 pandemic” (line 94) – consistency needed.

Lines 45-55: Readers who are unfamiliar with parkrun will benefit from a much broader explanation of the “5km event”. For example, it’s a free event, hosted in public open spaces. It is important to explain that people can run, walk, jog or participate as volunteers. Given the focus on new participants, it might be good to explain how people register and what they do (e.g. with their personal barcode/ID number) to get a record of their completion time, and how this feeds into the parkrun dataset.

Lines 59-62: Where was this study based?

Lines 63-65: More detail needed here to help justify the focus on parkrun. What does the previous research into parkrun say? How does parkrun improve mental health and wellbeing? You might want to consider citing the scoping review by Grunseit et al. here (cited later on in the manuscript).

Line 71: mention of health practitioners prescribing parkrun could be expanded upon- perhaps helping to justify why this research is focussing on parkrun as a public health offer.

Line 67: a rationale is provided for why the analysis looks at parkrun venues with blue space, but there is no rationale for why the analysis segregates by freshwater and saltwater. Could this make a difference?

Methods

General comments about Methods: The methods section would benefit from more detail in places. It needs an explanation of how “remoteness” was calculated and also how travel time to the parkrun was determined. Why was travel time preferred over distance? (see point below).

Line 87: adults - is this 18 years and over – please specify.

Line 94-95: the additional characteristics of parkrun venues – where was this information collected from? By who?

Line 103: I admit to being no expert in this analysis method. Is generalised linear mixed modelling the same as generalised linear modelling? If so, consistency needed throughout.

Line 110: how was “remoteness” of the event determined? This detail is needed in the methods. You may want to refer to previous studies by Robert Smith and Paul Schneider that have looked specifically at parkrun locations in the England.

Smith, R. A., Schneider, P. P., Cosulich, R., Quirk, H., Bullas, A. M., Haake, S. J., &; Goyder, E. (2021). Socioeconomic inequalities in distance to and participation in a community-based running and walking activity: A longitudinal ecological study of parkrun 2010 to 2019. Health & place, 71, 102626-102626. doi:10.1016/j.healthplace.2021.102626

Schneider, P., Smith, R. A., Bullas, A. M., Quirk, H., Bayley, T., Haake, S. J., . . . Goyder, E. (2020). Multiple deprivation and geographic distance to community physical activity events — achieving equitable access to parkrun in England. Public health (London), 189, 48-53. doi:10.1016/j.puhe.2020.09.002

Results

Lines 129-130: “The mean travelling time to the next nearest event was 30 mins for those that returned and 33 for those that did not return.” How was travelling time calculated? This detail is needed in the methods.

Line 158: Is this study only looking at “runners” or might some completion times be from those who walked (e.g., over 50-55mins might be consider walkers)? Be careful with terminology.

Line 168: “reasonable comparable to” 1) do you mean reasonably 2) it seems a bit vague to say “reasonably comparable” – is it comparable or not? Is this a meaningful comparison to make?

Discussion

General comments about Discussion: In the introduction, parkrun is used here as an example of a ‘positive upstream’ initiative that can have an important role in the obesity epidemic. I would like to see the Discussion come back to this point – how do the findings support this claim and what are the wider implications?

Line 181-182: “The wider gap in Scotland might be indicative of a larger gender gap in activity in Scotland”. Can you support this proposition with evidence/a citation?

Line 196: “lower” (travelling time) – might ‘shorter’ be a more appropriate word? I also want to know if this travelling time is by car, foot, as the crow flies or taking roads etc. Much more detail needed in Methods.

Implications for parkrun

There is a focus on parkrun (understandably, given the data), but I wonder if the authors could demonstrate any wider implications of the findings for other community initiatives like parkrun? This is important for readers from countries that do not have parkrun events, and would increase the relevance/impact of the findings.

Line 227: the paper would benefit from introducing the parkrun practice in the Introduction – this could be part of the rationale for focussing on parkrun as a public health initiative.

Limitations of the study

Line 236: “It is known that the gender gap in participation has been narrowing” – please specify, is this in the parkrun population? If so, provide a citation/reference.

6. PLOS authors have the option to publish the peer review history of their article (what does this mean?). If published, this will include your full peer review and any attached files.

**Do you want your identity to be public for this peer review?** For information about this choice, including consent withdrawal, please see our Privacy Policy.

Reviewer #1: **Yes: **Kate Black

Reviewer #2: No

---

## [Decision Letter · Decision Letter 1]

4 Jun 2023

PGPH-D-23-00368R1

Identification of novel characteristics that encourage first-time adult participants to return to parkrun in Scotland

Dear Dr. Gilburn,

Thank you for submitting your manuscript to PLOS Global Public Health. After careful consideration, we feel that it has merit but does not fully meet PLOS Global Public Health’s publication criteria as it currently stands. Therefore, we invite you to submit a revised version of the manuscript that addresses the points raised during the review process.

The revised manuscript has undergone evaluation by the two previous reviewers, and their comments are provided below. The reviewers expressed satisfaction with the manuscript's revision, which incorporated their suggestions, and acknowledged that the study's implications are now communicated more clearly.

However, the reviewers still have a few remaining concerns that would benefit from further attention.

We look forward to receiving your revised manuscript.

Kind regards,

Lucinda Shen, MSc

Staff Editor

Journal Requirements:

Additional Editor Comments (if provided):

Reviewers' comments:

Reviewer's Responses to Questions

**Comments to the Author**

1. If the authors have adequately addressed your comments raised in a previous round of review and you feel that this manuscript is now acceptable for publication, you may indicate that here to bypass the “Comments to the Author” section, enter your conflict of interest statement in the “Confidential to Editor” section, and submit your "Accept" recommendation.

Reviewer #1: All comments have been addressed

Reviewer #2: All comments have been addressed

2. Does this manuscript meet PLOS Global Public Health’s publication criteria? Is the manuscript technically sound, and do the data support the conclusions? The manuscript must describe methodologically and ethically rigorous research with conclusions that are appropriately drawn based on the data presented.

Reviewer #1: Yes

Reviewer #2: Yes

3. Has the statistical analysis been performed appropriately and rigorously?

Reviewer #1: I don't know

Reviewer #2: I don't know

4. Have the authors made all data underlying the findings in their manuscript fully available (please refer to the Data Availability Statement at the start of the manuscript PDF file)?

Reviewer #1: Yes

Reviewer #2: (No Response)

5. Is the manuscript presented in an intelligible fashion and written in standard English?

Reviewer #1: No

Reviewer #2: Yes

6. Review Comments to the Author

Reviewer #1: Thank you very much indeed for the opportunity to re-review your manuscript. It is evident that you have taken the majority of the reviewers’ comments on board in this revised manuscript. The methods that you have used to extract and analyse the data are now much clearer, as is the analysis itself (although I have only a relatively limited knowledge of quants analysis). Your findings are also much clearer as are the implications of your findings for parkrun’s future activity. Importantly, you have addressed one of my key concerns that was that in the initial manuscript you have seemingly identified only characteristics of parkruns that encourage return, from an unsubstantiated bounded set. That this is not the case is now far clearer. Thank you.

My main concerns remain as follows:

1. There is no overarching RQ but a series of RQs that while related necessitate the reader to make those connections. I would encourage you to consider identifying one over-arching RQ and then a series of sub-Qs (or perhaps even objectives?). I do wonder if a single focus upon geospatial features would make for a better paper?

2. You draw upon gender differences and barriers to exercise explicitly at the outset but then gender isn’t actually one of your questions / the focus of your paper (although it does then reappear so perhaps it does need to be included within the RQs?). I would encourage you to reconsider what you foreground in the introduction (see point 3)

3. I would encourage you to reconsider the structuring of the “Introduction” as at present there is no clear narrative through it that relates to and informs the overall purpose of your study. While I do recognise and appreciate that you have attempted to respond to the previous comments of the need to further recognise the extant literature, this has resulted in a somewhat amorphous section that is highly descriptive and needs greater synthesis. I do think all that you mention is relevant, it is just how it is structured and what is foregrounded / backgrounded that needs further reworking

4. Would not the pandemic have affected return rates by changing behaviours, and would this potentially not be more important than the variables observed? For example, new runners may have, in that intervening time period moved up from being a new/novice runner and have moved onto longer runs / additional activities?

5. Discussion: I would encourage you to stick to what you have found out and try to explain these findings rather than trying to make unsubstantiated ‘perhaps’ assertions. Ensure that you focus upon your RQs in this section

Minor points that you should also consider are:

1. Line 48 it is not clear why you are referring to “cycle routes” (as only an insider might recognise that parkruns are sometimes on cycle routes). Perhaps “traffic free routes” would be a better explanation at this point in the paper?

2. Line 64 “This is currently being promoted through the parkwalk” – what is? And walking what? – the route

3. Line 78 “In addition to the health benefits …” I think you mean physical health benefits as community, enhanced sense of self etc all relate to mental health – and therefore arguably are also ‘health benefits’ rather than ‘in addition to ..’

4. Line 319 you refer to “moving water” but surely coastal parkruns would have far more moving water than the other water-based locations? This (among some other assertion within) brings me to question your assertions re freshwater throughout the paper as the characteristics of it that you assert are very similar, if not more pertinent, for coastal locations as well. I think you need greater clarity here therefore in distinguishing these.

5. Lines 363-364, you suggest that more returners are female but I didn’t see this within the data that you presented

I look forward to reading your revised manuscript

Reviewer #2: All my comments have been addressed comprehensively. The revised manuscript is much improved. The description of parkrun provides much more detail for readers who are unfamiliar.

Given that much of the manuscript has been revised, I have some additional comments to make. These are just minor:

Line 75: should run be 'runs'

Line 98/99: 'performance of the parkrun population was falling' didn't quite sit right with me. Consider rephrasing. "the average finish time is getting longer" (again, I'm careful not to use the word slower, as I think that's a bit degrading!)

Line 105: is there scope to briefly add what some of the major barriers to participation were here?

Line 179: KML format- does this need writing in full?

Line 290-292: repetition of 'relatively slow' in this sentence. Is there any evidence/citation to support this claim of slower participants feeling like they're holding up the event? (I'm wondering if any of the barriers to parkrun research has this evidence?)

line 343: check consistency of using hyphen in 'first-time' throughout manuscript

I've enjoyed reading this paper and hope the readers do too.

Please note, the reason I have selected "I don't know" about the appropriateness and rigour of statistical analysis is because I have no experience with the analysis used in this paper.

7. PLOS authors have the option to publish the peer review history of their article (what does this mean?). If published, this will include your full peer review and any attached files.

**Do you want your identity to be public for this peer review?** For information about this choice, including consent withdrawal, please see our Privacy Policy.

Reviewer #1: No

Reviewer #2: **Yes: **Helen Quirk

---

## [Decision Letter · Decision Letter 2]

10 Jul 2023

PGPH-D-23-00368R2

Identification of novel characteristics associated with first-time adult participants returning to parkrun in Scotland

Dear Dr. Gilburn,

Thank you for submitting your manuscript to PLOS Global Public Health. After careful consideration, we feel that it has merit but does not fully meet PLOS Global Public Health’s publication criteria as it currently stands. Therefore, we invite you to submit a revised version of the manuscript that addresses the points raised during the review process.

EDITOR'S COMMENT: 

An observation is made regarding the benefit of minor revision to the title of the manuscript is suggested.  Kindly consider this.

We look forward to receiving your revised manuscript.

Kind regards,

Nnodimele Onuigbo Atulomah, PhD

Academic Editor

Journal Requirements:

Additional Editor Comments (if provided):

Having reviewed the outcomes of the rounds of reviews submitted, it is believed that the manuscript would greatly benefit from a title modification from what it is currently to "Predictors of successful return to Parkrun among first-time adults in Scotland" considering that the short title is "Factors Associated With Return Rates to parkrun" and harmonizes appropriately with the suggested revised title and the data analysis in the manuscript and findings. With this revision the manuscript is well poised to communicate adequately with potential readers. If this suggestion is accepted, kindly replace "novel characteristics" and modify the statement of the aim to align appropriately with the revised title.

Reviewers' comments:

Reviewer's Responses to Questions

**Comments to the Author**

1. If the authors have adequately addressed your comments raised in a previous round of review and you feel that this manuscript is now acceptable for publication, you may indicate that here to bypass the “Comments to the Author” section, enter your conflict of interest statement in the “Confidential to Editor” section, and submit your "Accept" recommendation.

Reviewer #1: All comments have been addressed

Reviewer #2: All comments have been addressed

2. Does this manuscript meet PLOS Global Public Health’s publication criteria? Is the manuscript technically sound, and do the data support the conclusions? The manuscript must describe methodologically and ethically rigorous research with conclusions that are appropriately drawn based on the data presented.

Reviewer #1: Yes

Reviewer #2: Yes

3. Has the statistical analysis been performed appropriately and rigorously?

Reviewer #1: I don't know

Reviewer #2: I don't know

4. Have the authors made all data underlying the findings in their manuscript fully available (please refer to the Data Availability Statement at the start of the manuscript PDF file)?

Reviewer #1: Yes

Reviewer #2: Yes

5. Is the manuscript presented in an intelligible fashion and written in standard English?

Reviewer #1: Yes

Reviewer #2: Yes

6. Review Comments to the Author

Reviewer #1: Thank you very much indeed for the opportunity to further re-review your manuscript. I appreciate you responding to many of the comments in this revised manuscript. In particular, thank you for ensuring that the study aim, and associated questions, are now clearly stated and that the narrative through the paper is much more apparent. Thank you.

My suggestions are as follows:

1. Your work would have greater initial impact, and understanding, if you direct the reader towards parkrun earlier in your initial statements around exercise, by drawing upon parkrun-relevant examples – for example that exercising with others can be really important … rather than e.g. provision of off road routeways!

2. That many other quants studies have taken place isn’t in itself a reason to undertake the work! Are you better able to justify the study?

3. Some of the variables that you state within the RQs aren’t entirely evident in your preceding literature review

4. As previously, research has shown that the pandemic has significantly changed individuals’ behaviours, and especially towards exercise. Therefore, I would encourage you to consider would not the pandemic have affected return rates by changing behaviours, and would this potentially not be more important than the variables observed? For example, new runners may have, in that intervening time period moved up from being a new/novice runner and have moved onto longer runs / additional activities? You acknowledge this in your comments but I would have expect to see some recognition of this in the “limitations”

Reviewer #2: All comments have been addressed.

I have no further comments.

Please note, the reason I have selected "I don't know" about the appropriateness and rigour of statistical analysis is because I have no experience with the analysis used in this paper.

7. PLOS authors have the option to publish the peer review history of their article (what does this mean?). If published, this will include your full peer review and any attached files.

**Do you want your identity to be public for this peer review?** For information about this choice, including consent withdrawal, please see our Privacy Policy.

Reviewer #1: No

Reviewer #2: No

---

## [Editor Report · Decision Letter 3]

21 Jul 2023

Predictors of successful return to parkrun for first-time adult participants in Scotland

PGPH-D-23-00368R3

Dear Mr. Gilburn,

We are pleased to inform you that your manuscript 'Predictors of successful return to parkrun for first-time adult participants in Scotland' has been provisionally accepted for publication in PLOS Global Public Health.

Best regards,

Nnodimele Onuigbo Atulomah, PhD

Academic Editor

Congratulations for having completed the rounds of reviews and revisions successfully.